# Sprinters’ and Marathon Runners’ Performances Are Better Explained by Muscle Fibers’ Percentage Cross-Sectional Area than Any Other Parameter of Muscle Fiber Composition

**DOI:** 10.3390/sports13030074

**Published:** 2025-03-04

**Authors:** Spyridon Methenitis, Angeliki-Nikoletta Stasinaki, Thomas Mpampoulis, Constantinos Papadopoulos, Giorgos Papadimas, Nikolaos Zaras, Gerasimos Terzis

**Affiliations:** 1Sports Performance Laboratory, School of Physical Education & Sports Science, National and Kapodistrian University of Athens, 17237 Athens, Greece; agstasin@phed.uoa.gr (A.-N.S.); thompamp@phed.uoa.gr (T.M.); 2Human Body Analysis Lab, Theseus, Physical Medicine and Rehabilitation Center, 17672 Athens, Greece; 3A’ Neurology Clinic, Aiginition Hospital, Medical School, National and Kapodistrian University of Athens, 11528 Athens, Greece; constantinospapadopoulos@yahoo.com (C.P.); gkpapad@yahoo.gr (G.P.); 4Department of Physical Education and Sport Science, School of Physical Education, Sport Science and Occupational Therapy, Democritus University of Thrace, 69100 Komotini, Greece; nzaras@phyed.duth.gr

**Keywords:** muscle fiber type, sprinting, jumping, strength, power, rate of force development, maximum oxygen consumption, velocity at maximum oxygen consumption

## Abstract

The present study aimed to investigate the correlation between muscle fiber type variables and sprinting, jumping, strength, power and endurance performances in sprinters and marathon runners. Furthermore, the study explored which muscle fiber type variable influences athletes’ performance the most and which variable discriminates sprinters from marathon runners. Body composition, vastus lateralis muscle fiber composition, sprinting, jumping, strength, maximum isometric force, rate of force development and endurance performance were assessed in well-trained sprinters (n = 15; age: 24.7 ± 4.1 years; body mass: 82.7 ± 8.8 kg; body fat: 11.3 ± 4.2%) and marathon runners (n = 15; age: 26.1 ± 3.2 years; body mass: 69.7 ± 4.3 kg; body fat: 9.5 ± 3.3%). Significant correlations were observed between muscle fiber composition variables and performances (r = −0.848 to 0.902; *p* < 0.05), with percentage cross-sectional areas (%CSAs) showing the strongest associations. Regression analyses confirmed %CSAs as the strongest predictors of performance (R^2^: 0.796–0.978; *p* < 0.001; B coefficients: −0.293–0.985), while discriminant analysis accurately differentiated sprinters from marathon runners (100%, *p* < 0.001) by only using athletes’ muscle fibers’ %CSAs. In conclusion, muscle fibers’ %CSAs may be the most critical variables for explaining and distinguishing sprinters’ and marathon runners’ performances.

## 1. Introduction

From the pioneer studies in the 1960s until now, the role of muscle fiber composition in human performance has been extensively studied. Several studies have explored the possible correlations between muscle fiber type distribution and endurance, sprinting, strength and power performances [1,2,3,4,5,6,7,8,9,10,11,12,13,14,15,16,17,18,19,20,21,22,23]. According to these reports, the correlations between sprinting, jumping, strength, power performance and percentages of Type I muscle fibers ranged from non-significant to significant, from negative to positive (r-Pearson ranged from −0.4 to 0.2), while for Type II, they ranged from non-significant to r = 0.8. For endurance performance, moderate to strong correlations (r-Pearson ranged from 0.4 to 0.8) have been reported for Type I, while for Type II, the correlations are conflicting, as non-significant to significant, negative to positive and low to moderate correlations have been reported [1,2,3,4,5,6,7,8,9,10,11,12,13,14,15,16,17,18,19,20,21,22]. Moreover, in many cases, the correlations observed between muscle fiber distribution and performance have been misunderstood and/or misused. For example, it is commonly reported that Type II muscle fibers are highly corelated with sprinting performance. However, these correlations are consistently observed when sprinting performance is evaluated through cycling and not when running sprinting performance is evaluated. Until now, moderate correlations have been reported between Type II muscle fibers’ percentage and sprinting performance only in well-trained sprinters [24], runners [25], as well as between Type IIa muscle fibers’ percentage and 10 m sprint performance in trained young soccer players [26]. Cycling and running are two different types of movements, with significantly different mechanical, physiological and metabolic demands [27,28]; therefore, the observed correlations in cycling could not be transferred to running performance. Thus, it is still debatable whether running sprinting performance may correlate with muscle fibers’ Type II percentages. Furthermore, even when significant correlations between muscle fiber type distribution and endurance, sprinting, strength, power performances are reported, these correlations can only explain a portion of the variations in participants’ performances, ranging between 10% and 64% [1,2,3,4,5,6,7,8,9,10,11,12,13,14,15,16,17,18,19,20,21,22]. Finally, it is commonly reported that endurance athletes are characterized by increased distribution of Type I muscle fibers, while power athletes (like sprinters, throwers and jumpers) are characterized by increased proportion and sizes of Type II muscle fibers [2,3,4,7,8,12,13,14,15,17]. However, there are several studies reporting that the percentages of each muscle fiber type alone cannot distinguish athletes of different sports and elite level from moderate level athletes [3,4,6,8,11,13,15,17,18,29,30]. According to these reports, athletes with the same endurance, sprinting, strength and power performances have different muscle fiber type distribution, or athletes with the same muscle fiber type distribution have significantly different endurance, sprinting, strength, power performances. Indeed, from the pioneer study by Horwitz, Sidossis and Coyle [6], it is known that athletes with significantly different muscle fiber distributions (one group possessed ~73.3% Type I muscle fibers, and the other group possessed ~48.2%) did not differ in terms of their VO_2max_ and VO_2_ at lactate threshold. Considering all of the above, even if the role of muscle fiber composition in human body performance has been extensively studied, the controversial results limit the safe conclusions about this topic, and they do not allow us to understand whether the different muscle fiber percentages that are consistently observed between athletes of different events are one of the major determinant parameters of their performance or whether athletes’ performance or categories can be distinguished by their muscle fiber type composition.

Several reasons can be identified for these controversial results, including differences among the participants in each study (the level of their fitness status, age, training background, etc.) and the different methodological approaches between studies. To date, several studies in this area have explored the possible correlations between all Type II muscle fibers or Type II myosin heavy chains (MHCs), including Type IIa and Type IIx as one variable, and performance, even if the two subgroups of Type II muscle fibers/MHC II have significantly different physiological, mechanical and metabolic properties [31,32,33,34,35,36,37], with Type IIa muscle fibers being closer to Type I than to Type IIx muscle fibers [31,32,33,34,35,36,37,38,39,40]. This is more often in studies investigating the possible correlations between endurance performance and muscle fiber distribution, in which, the majority of the studies, reported significant correlations between the percentage of Type I muscle fibers, but as they explore these correlations for all Type II muscle fibers as one variable, the reported correlations are either low or non-significant [6,7,9,10,12,16,30]. Thus, the role of Type IIa muscle fibers in endurance performance is still uncertain. Furthermore, the vast majority of studies in this area do not consider the size of muscle fibers (i.e., the cross-sectional area; CSA) when they explore the above relationships. The mechanical, physiological and metabolic properties of muscles are not determined only by the distribution of each muscle fiber type within them but also by the size of muscle fibers [31,32,33,34,35,36,37] and the specific training-induced adaptations on them [40,41,42,43,44,45]. For example, it has previously been reported that a world champion shot putter had a predominance of Type I muscle fiber in his vastus lateralis (Type I: 60%, Type II: 40%) [13], a fiber type distribution, which is excepted to be found in endurance athletes [2,3,6,7,8,9,10,12,18]. However, due to the very specific training that this shot putter had followed for years, the size of his Type II muscle fibers was almost 3 times larger than his Type I muscle fibers, and thus, his fewer Type II muscle fibers occupied almost 70% of his whole muscle area [13]. Considering this observation, as well as the fact that the mechanical, physiological and metabolic properties of muscle fibers are determined also by their size [31,32,33,34,35,36,37], it could be concluded that the increased area of muscle occupied by Type II muscle fibers that this shot putter had made his muscle behave as a muscle with Type II muscle fiber predominance and not as a muscle with Type I muscle fiber predominance. This observation further supports the opinion that perhaps it is the area of muscle that is occupied by each muscle fiber type (i.e., the percentage cross-sectional area; %CSA) that is more important and not either the percentage or the size of each muscle fiber type alone. Indeed, it seems that stronger correlations exist between %CSA of each muscle fiber type and strength/power performance, body composition, glycemic/lipidemic profiles in sedentary, recreational trained individuals, well-trained marathon, power and strength athletes [2,3,4,46,47], with the impact of each muscle fiber type %CSA on performances changing according to the individuals’ training experience and type of training, with the performances of more experienced, well-trained athletes relying more on their muscle fibers’ %CSAs compared to novice or untrained individuals [4]. As muscle fibers and muscles are characterized by high plasticity (the ability to adapt and change depending on the imposed physiological, mechanical and metabolic demands in each period), their morphological characteristics, muscle fiber distribution, cross-sectional area, physiological, metabolic and mechanical properties are not stable but change according to the imposed demands, with systematic training being one of the most important regulators of muscle fibers’ and muscles’ phenotypes [40,41,42,43,44,45]. The epigenetic, training-induced changes in muscle fibers and whole muscle phenotype seem to be better explained by each individual’s/athlete’s muscle fibers’ %CSA at a given time point compared to any other parameter of muscle fiber composition, as previously described [2,3,13]. Indeed, it seems that the training-induced adaptations in strength, power performance, body composition, glycemic/lipidemic profiles are better explained by the training-induced adaptations in muscle fibers’ %CSAs [48,49]. However, even for the %CSA, the results are still controversial. For example, Type I %CSA seems to be strongly related to maximum isometric strength in power athletes but not in endurance athletes, while even in power athletes, the magnitude and the significance of these correlations seem to change depending on the examined exercise [11]. Indeed, even in more recent studies, it seems that the reported correlations between muscle fibers’ %CSAs and endurance, strength and power performances are not constant, ranging from non-significant to very high [2,3,4,48]. Thus, the role of muscle fibers’ %CSAs in sprinting, jumping, strength, power and endurance performances is still debatable, while, at least to our knowledge, whether the %CSA may discriminate sprinters from marathon runners is still unknown.

It is important to identify which parameter of muscle fiber composition is stronger and consistently linked to athletes’ performance. Finally, it would be interesting to explore which parameter of muscle fiber composition could better be used to discriminate athletes of different sports or athletes with different training backgrounds. Unfortunately, to date, studies in this area have taken into account only one or two of the muscle fiber composition parameters, leaving a significant research gap in the literature. Therefore, the aim of the study was to investigate the correlation between muscle fiber type variables and sprinting, jumping, strength, power and endurance performances in sprinters and marathon runners. Furthermore, the study aimed to explore which parameter influences athletes’ performance the most and which parameter discriminates sprinters from marathon runners. It was hypothesized that the %CSA of each muscle fiber type may be the main predictor.

## 2. Materials and Methods

### 2.1. Experimental Approach

Athletes were recruited via advertisements in local sport clubs. Respondents visited the laboratory and completed a recall self-reported physical activity questionnaire [48]. Those who fulfilled the inclusion criteria visited the laboratory for a second time for their medical examination, anthropometric, body composition and lower extremity dominance evaluations [Waterloo Footedness Questionnaire; WFQ-R, Intraclass Correlation Coefficient (ICC) = 0.92; [2]] and signed an informed consent form. On their third visit, an evaluation of their maximum aerobic capacity was performed. One week later (4th visit), sprinting, jumping and leg press maximum isometric force and rate of force development performances were performed with a randomized design with 10 min rest between them. All athletes were familiar with all of the above performance evaluations. One week later, muscle biopsies from the vastus lateralis of each athlete’s dominant leg were obtained. All procedures were in compliance with the Declaration of Helsinki as revised in 2000 and were approved by the Ethics Committee of the School of Physical Education & Sports Science, National and Kapodistrian University of Athens (number 1039/14-02-2018).

### 2.2. Participants

Thirty young males volunteered to participate in the present study; they were assigned into two groups depending on their training background: Sprinters (SP; n = 15; Table 1), including eight 100 m (best performance 10:88 ± 0.25 s) and seven 200 m sprinters (best performance 21:01 ± 0.26 s), and Marathon Runners (MR; n = 15; Table 1; best performance 2 h 45 min ± 17.15 min). All athletes ranked between the second and fifteenth place at the previous national outdoor championships. The inclusion criteria were (1) at least five years of systematic training for their event, (2) weight stability (±2 kg) prior to entry (~3 months), (3) absence of restraining orthopedic and neuromuscular maladies, (4) aged between 18 and 35 years, (5) absence of drug abuse or medications, which are known to affect the neuromuscular system and performance, (6) participation in the most recent national championships.

### 2.3. Procedures

#### 2.3.1. Body Composition

Body height was measured using a stadiometer, with an accuracy of 0.5 cm (SECA 220, Seca Corporation, Columbia, SC, USA). Total body scan was performed using dual energy X-ray absorptiometry (DPX-L; LUNAR Radiation, Madison, WI, USA) during morning hours (8–10 a.m.) and after overnight fasting. All measurements were analyzed using the LUNAR radiation body composition program. Body mass, fat mass (% and kg) and lean body mass (LBM) were determined for total body as well as for lower extremities. The intraclass correlation coefficients (ICCs; Intrarater) for body composition variables were as follows: lower extremities LBM = 0.98 (95% CI: Lower = 0.95, Upper = 0.99), total LBM = 0.93 (95% CI: Lower = 0.89, Upper = 0.97), total % fat = 0.90 (95% CI: Lower = 0.85, Upper = 0.96) and lower extremities % fat = 0.94 (95% CI: Lower = 0.88, Upper = 0.98) (*p* < 0.0001).

#### 2.3.2. Sprinting Performance

Sprint running performance, i.e., sprinting time (10 m and 60 m), was measured outdoors on a track surface at an ambient temperature between 19 and 23 °C and wind velocity between 0 and +0.8 m∙s^−1^, as previously described [4]. Three pairs of wireless photocells (Brower Timing System, Draper, UT, USA; accuracy 0.01 s) were placed at 0 m, 10 m and 60 m. Athletes initially performed three submaximal sprints, with 3 min of rest between them, and after 5 min, they performed three maximal sprints (standing start position) separated by 5 min rest intervals [4]. The best performance was retained for statistics (ICC = 0.91; 95% CI: Lower = 0.86, Upper = 0.95). Mean velocity was calculated accordingly: Distance · time^−1^.

#### 2.3.3. Jumping Performance

Jumping performance was evaluated through counter movement jumps (CMJs) on a force platform (Applied Measurements Co., Ltd., Aldermaston, UK, WP800–1000 kg weighting platform, s/n: 40245, 80 cm × 80, sampling frequency 1 kHz) with arms akimbo, as previously described [2,48]. Initially, each athlete performed 3 CMJs with submaximal intensity, followed by 3 maximal CMJs with 2 min rest between each attempt [2,48]. Data from the force platform were recorded and analyzed (Kyowa sensor interface PCD-320A) in order to calculate the following variables: [Jump height (cm) = ((0.5 ∙ flight time)^2^ ∙ 2^−1^) ∙ 9.81] and [Maximum power (W) = (body weight + Fmax) ∙ 9.81 ∙ flight time]. The signal was filtered using a secondary low-pass Butterworth filter with a cutoff frequency of 10 Hz. The best performance according to the jump height was used for further analysis. The ICCs for jump height and power were 0.87 (95% CI: Lower = 0.83, Upper = 0.95) and 0.91 (95% CI: Lower = 0.90, Upper = 0.99), respectively.

#### 2.3.4. Leg Press Isometric Force and Rate of Force Development

Athletes were seated on a custom-made steel leg press chair and placed both their feet on the force platform (Applied Measurements Ltd. Co., UK, WP800, 1000 kg weighting platform, 80 × 80 cm, sampling frequency 1000 Hz), which was positioned perpendicular on a concrete laboratory wall. The knee angle was set at 120°, and the hip angle was set at 100°, as previously described [2,48]. All athletes were instructed to apply their maximum force as fast as possible for 5 s. Five submaximal efforts were performed before maximal tests. Three maximum trials were performed with 2 min rest intervals between them. Athletes were vocally encouraged to perform their best. Real-time visual feedback of the force applied was provided for each effort via a computer monitor placed just above the force platform. Variables calculated from the force–time curve included maximum isometric force (MIF) and the rate of force development (RFD), as previously described [2,48,50]. MIF was determined as the highest peak on the curve, and RFDs at 60, 120 and 250 ms from the onset of muscle contraction were calculated according to the following equation: RFD (N∙s^−1^) = ΔForce ∙ ΔTime^−1^. The best performance according to RFD at 120 ms was used for statistical analysis. The ICCs for MIF and RFD were 0.90 (95% CI: Lower = 0.86, Upper = 0.96) and 0.92 (95% CI: Lower = 0.80, Upper = 0.98), respectively.

#### 2.3.5. Maximum Aerobic Capacity

An incremental protocol [51,52] on a motor-driven treadmill (Technogym Runrace, Gambettola, Cesena, FC, Italy) until exhaustion was used. After 10 min of a standard warm-up and 5 min of passive rest, athletes started their effort on the treadmill. Initially, the treadmill speed was set at 10 km∙h^−1^ and 1% inclination. The speed was increased by 1 km∙h^−1^ every minute until exhaustion [51,52]. Heart rate was continuously monitored via telemetry (Polar FT1, Polar Electro, Kempele, Finland), and perceived exertion (Borg scale) was recorded every minute. Unfortunately, due to equipment failure, after evaluation of the first 6 athletes, maximum oxygen consumption was estimated based on the maximal heart rate of each athlete achieved at the end of the test using the following equation [53]: VO_2peak_ (mL∙kg^−1^∙min^−1^): (max heart rate ∙ heart rate rest^−1^) ∙ 15 (the ICC for VO_2peak_ was 0.87; 95% CI: Lower = 0.81, Upper = 0.94). The running speed and heart rate at the last completed stage were determined as maximum aerobic velocity (MAV) and maximum heart rate (MHR), respectively.

#### 2.3.6. Muscle Biopsies and Histochemistry

Muscle samples were obtained using Bergström needles (Bergström Muscle Biopsy Cannula, 5 mm diameter, STILLE, IMEMA, Athens, Greece) from the middle part of the non-dominant lower extremity vastus lateralis under local anesthesia by a trained medical doctor. Samples were aligned, placed in an embedding compound and frozen in isopentane pre-cooled to its freezing point, and they were subsequently stored in liquid nitrogen until analysis. Serial 10 μm thick cross-sections were cut at −20 °C and stained for myofibrillar ATPase after pre-incubation at pH 4.3, 4.6 and 10.3 [54]. A mean of 762 ± 233 muscle fibers from each athlete were classified as type I, IIa or IIx. The CSA and %CSA of all the classified muscle fibers were measured with an image analysis system (Image Pro, Media Cybemetics Inc., Silver Spring, MD, USA) at a known and calibrated magnification [46,47]. The ICCs for these evaluations in our laboratory ranged from 0.93 to 0.96 [95% CI: Lower = 0.91, 0.92, 0.87 and Upper = 0.99, 0.98, 0.95, respectively].

### 2.4. Statistical Analyses

A post hoc power analysis was used for the determination of the results’ actual power (G*Power ver. 3.1; Frank Faul, Universitat Kiel, Germany). The power for the correlation coefficients when all participants were included (n = 30) ranged between 0.875 and 0.902, while for all the other analyses, it ranged between 0.850 and 0.898. All data are presented as means and standard deviation (±SD). The Shapiro–Wilk test was used to check the normality of the data. Νo violations were found. Independent sample *T*-test was used for the determination of differences between the groups. Pearson’s product-moment correlation coefficient analysis was used to explore the relationships between the evaluated parameters of muscle fiber composition and athletes’ performance measurements. The interpretation of the observed correlations was performed according to Hopkins’ ranking: correlation coefficients between 0.3 and 0.5 were considered moderate, between 0.51 and 0.70 large, between 0.71 and 0.90 very large and >0.91 almost perfect [55]. Adjusted R square was also calculated for all significant correlations to explore the explanation proportion of each correlation. Partial correlations were used to investigate the possible effect of lean mass on the above correlations. Multiple regression analyses (stepwise) were used for the determination of the best linear combination, as well as of the muscle fibers’ composition parameters’ relative strengths, for the explanation of athletes’ performances. A linear discriminant analysis (LDA) was performed to identify which combination of the examined muscle fibers’ composition parameters could best discriminate Sprinters from Marathon Runners. The discriminant function was derived by maximizing the ratio of between-group variance to within-group variance. The analysis generated group centroids, canonical discriminant functions and a set of discriminant coefficients for each variable. The significance of each discriminant function was assessed using Wilks’ Lambda, with a significance threshold set at *p* < 0.05. Additionally, the eigenvalues for each discriminant function were computed to quantify the amount of variance explained by each function. To assess the performance of the discriminant model, classification accuracy was calculated based on the number of correct classifications in the test dataset. A confusion matrix was used to evaluate the model’s ability to correctly classify individuals into the two groups. The overall classification accuracy was calculated as a percentage of correctly classified cases relative to the total number of observations. Additionally, the canonical discriminant functions were interpreted to determine the importance of each predictor variable in distinguishing between the groups. Discriminant coefficients greater than 0.300 were considered to indicate an important contribution, while those greater than 0.5 were considered to indicate a strong contribution to the separation of the groups. Statistical analyses were performed using SPSS Statistics Ver. 20 (IBM Corporation, SPSS Inc., Chicago, IL, USA). *p* ≤ 0.05 was used as a two-tailed level of significance.

## 3. Results

Significant differences were found between the groups for body mass, total and lower extremities lean body mass, percentage of type I, IIx, II muscle fibers, CSA of type IIa and IIx muscle fibers, %CSA of all muscle fiber types and in all parameters of performance, which were evaluated in the present study (*p* < 0.05; Table 1). The percentage distribution of each muscle fiber type did not differ between groups (*p* > 0.152; Table 1).

Significantly large correlations were observed between the CSA of muscle fibers Type I, IIa and IIx (r: −0.666–0.700; range of variation explained by each correlation: 44.3–49.0%) and sprinting–jumping performances (*p* < 0.05; Table 2). In contrast, very large to almost perfect correlations were observed between athletes’ sprinting, jumping performance (*p* < 0.005; Table 2) and the %CSA of muscle occupied by Type I (r: −0.848–0.842; variation explained: 69.9–71.9%), Type IIa (r: −0.811–0.869; variation explained: 65.7–75.5%) and Type IIx (r: −0.895–0.892; variation explained: 79.5–80.1%). No significant correlations were found between sprinting performances and the percentage of each muscle fiber type (*p* > 0.215), while moderate correlations were observed between jumping performance and muscle fiber distribution (r: −0.481–0.589; variation explained: 23.1–34.7%; Table 2).

Leg press MIF and RFD (60 ms, 120 ms, 250 ms from the onset of muscle contraction) were moderately correlated with the percentages of muscle fiber types (r: −0.514–0.528; variation explained: 26.4–27.8%) and moderately to largely correlated with the CSA of muscle fibers (r: −0.771–0.902; variation explained: 59.4–81.3%). Moderate correlations were observed between estimated VO_2max_, MAV and Type I, IIx muscle fibers’ percentages (r: −0.502–0.505; *p* < 0.05; variation explained: 25.2–25.5%). No significant correlations were observed for the percentage of Type IIa muscle fibers or for the CSA of all muscle fibers (*p* > 0.05; Table 2). Large correlations were found between the %CSA of muscle occupied by Type I muscle fibers (r: 0.875–0.890; variation explained: 76.5–79.2%), Type IIx muscle fibers (r: −0.824–−0.814; variation explained: 66.2–67.9%) and VO_2max_, MAV. When all athletes were examined as one group, the %CSA of muscle occupied by Type IIa muscle fibers did not correlate either with VO_2max_ or MAV. This was a surprise for us because significant correlations between the %CSA of Type IIa muscle fibers and all previous performance parameters were consistently observed. Thus, we further examined these observations by investigating the possible correlations between the %CSA of Type IIa muscle fibers and VO_2max_ and MAV in each group separately. When we performed this analysis for each group separately, the %CSA of Type IIa muscle fibers demonstrated large correlations with the estimated VO_2max_ and MAV in Sprinters (r: 0.731–0.853; variation explained: 53.4–72.9%; Figure 1) and Marathon Runners (r: 0.750–0.760; variation explained: 56.3–57.8%; Figure 1). Considering the above observation, we performed all correlations for each group separately; however, the magnitude of the correlations in each group separately did not differ from those observed when all participants were examined as one group (see Figure 1). Hence, Table 1 presents only significant correlations for all participants as one group. Finally, when the above correlations were controlled for the possible effect of athletes’ lean mass, the magnitude of the correlations was not affected significantly (range: −0825 to 0.895).

Multiple regression analyses were used to investigate which variables of muscle fiber composition examined in the present study could better explain (relative strength) the athletes’ performance. The results for the determination of sprinting, jumping, MIF and RFD, for which linear regression analyses were performed for all athletes as one group, are analytically presented in Table 3, while, based on the previously presented observation, for the determination of athletes’ endurance performance, linear regression analyses were performed for each group separately, and the results are presented in Table 4. In short, the athletes’ performance could be better explained by the linear combination of %CSAs of muscle fibers (B coefficients ranged between −0.849 and 0.985) and not by the percentages or CSAs of muscle fibers (Table 3 and Table 4).

LDA was used to identify which combination of the examined muscle fibers’ composition parameters (%, CSA and %CSA) could best discriminate Sprinters from Marathon Runners. The analysis revealed one highly significant discriminant function (eigenvalue: 14.318; % of variance: 100%; canonical correlation: 0.967; Wilks’ Lambda: 0.065; X^2^: 53.216; *p* < 0.001). The centroids of the groups were clearly separated (Sprinters: −3.113; Marathon Runners: 4.245). Accordingly, the LDA model correctly classified 100.0% of originally grouped cases and 100.0% of cross-validated grouped cases as either Sprinters or Marathon Runners. Table 5 presents the canonical discriminant function coefficients, Wilks’ Lambda and *p* values of each muscle fiber composition parameter that was used for the development of the LDA function. The key contributors, in order of their relative impact, were the %CSA of Type IIx, Type I and Type IIa muscle fibers (Table 5). Although Type I percentage, the CSA of Type I and IIa muscle fibers were significant, their contributions in the LDA function were very weak, falling below the 0.300 threshold; therefore, they were not considered to be important contributors. No significant contributions were observed for the percentage of Type IIa, IIx and the CSA of Type IIx muscle fibers (*p* > 0.121; Table 5).

## 4. Discussion

The main findings of the present study are that the percentage cross-sectional area of the vastus lateralis muscle occupied by Type I, IIa and IIx muscle fibers seems to be the determinant parameter that can better explain the differences and the performances between Sprinters and Marathon Runners and not the percentage or the cross-sectional area of each muscle fiber type that these athletes possess. Similarly, the %CSA of muscle occupied by Type I, IIa and IIx muscle fibers was the only parameter that could discriminate Sprinters from Marathon Runners and not any other variable of muscle fiber composition.

In the present study, as expected, the percentages of each muscle fiber type did not correlate with sprinting performance (running)—either when Sprinters and Marathon Runners were controlled for as one group or when these correlations were explored for each group separately. To date, only a handful of studies have reported significant correlations between muscle fiber percentages and sprint running performances [24,25,26], while the vast majority of studies in this area have found no significant correlations. In contrast, significant but moderate and type-dependent correlations were observed between muscle fiber Type I, IIa and IIx percentages and jumping, MIF, RFD_250ms_ and endurance performance, which, however, could only explain a small portion of the variations in each performance, ranging between 22.6% and 34.9%. The observed correlations are not a new phenomenon, as they have been reported many times to date [1,2,3,4,5,6,7,8,9,10,11,12,13,14,15,16,17,18,19,20,21,22,48]. Even in these reports, the observed correlations between muscle fiber distribution and jumping, strength, power and endurance performances could only explain a small portion of the variations. Furthermore, in several of the above-mentioned studies, the significant positive or negative relationship (type-dependent) that was observed seemed to be indirect or affected by other biological factors [1,2,3,4,5,6,7,8,9,10,11,12,13,14,15,16,17,18,19,20,21,22,48]. For example, in a study by Ivy et al. [10], significant correlations were found between maximum oxygen consumption and the percentage of Type I muscle fibers; however, when the effect of participants’ respiratory capacity was removed, the above correlation became insignificant. In support of the above report, it seems that the observed low to moderate correlations between the percentages of muscle fibers and endurance, strength and power performances are dictated by the metabolic and mechanical properties of muscle fibers, as when the observed correlations were controlled for the effect of metabolic and mechanical properties of muscle fibers, the relationships again became insignificant [5,6,7,9,10,11,12,14,18,25,30,35]. Indeed, it seems that only when a combination of muscle fiber percentages and other biological factors, such as respiratory capacity, metabolic and mechanical properties of muscle fibers, etc., are used can human body performance be better explained [8,10,12,16,29,56]. Furthermore, even when significant differences were observed between the two groups in the present study, as for the percentages of Type I, IIx, II muscle fibers and in all evaluated performance parameters within each group, athletes demonstrated a wider range of each muscle fiber type percentages, which indicates that athletes from the same group with similar performance in a physical evaluation may have significantly different muscle fiber distribution or vice versa. Accordingly, the percentage of each muscle fiber type does not seem to be able to provide all the necessary information that could be used for the explanation of athletes’ performance, even within a group of Sprinters or Marathon Runners. Thus, by examining only the number/percentages of each muscle fiber type within a muscle, crucial information regarding metabolic and mechanical properties of the muscle or how the muscle works is missing. Indeed, muscle fibers’ metabolic and mechanical properties are significantly affected not only by their type but also by the size of each muscle fiber, with larger muscle fibers having different metabolic and mechanical properties compared to smaller muscle fibers, even when they belong to the same type [2,6,8,13,14,15,18,21,29,31,32,33,34,35,36,37,40,44,46,47,50,57,58]. In addition, all muscles are affected by the type of systematic training, with different types of training inducing significant training-type-dependent adaptations on them [14,40,41,42,43,44,45,57] by affecting gene expression and protein synthesis and thus altering the content of specific muscle proteins within muscle fibers, which in turn alters the size but mostly the metabolic and mechanical properties of each muscle fiber, even one of the same type, and consequently, the way each muscle fiber acts and behaves during any muscle contraction [14,40,41,42,43,44,45,58,59,60]. Indeed, it has been reported that large/trained Type I muscle fibers seem to produce similar force/power to small/untrained Type II muscle fibers or that large/trained Type IIa muscle fibers have similar metabolic properties to small/untrained Type I muscle fibers, especially during dynamic moderate to fast contractions [14,16,18,39,57,61,62]. All of the above, along with findings of the present study, suggests that the percentage of muscle fibers is neither the primary determinant of human physical performance nor sufficient to provide the necessary insights to fully understand athletes’ performance. This conclusion is further supported by results of the multiple regression analysis and LDA performed in the present study, in which the percentages of each muscle fiber type of these athletes did not have any significant relative strength in the determination/explanation of athletes’ performance, and they could not discriminate Sprinters from Marathon Runners. Considering all of the above, it could be concluded that the percentages of each muscle fiber type are not the determinant parameter that could strongly explain or affect these performances, even in well-trained athletes, nor could they be used to discriminate between athletes from different sports.

Higher correlations, compared to those observed for the percentages, were found between the cross-sectional areas (CSAs) of all muscle fiber types and strength/power performances. The correlations were fiber-type-dependent. Furthermore, significant correlations were observed between the CSA of Type IIa, IIx and sprinting performance. To date, it has been well described that sprinting, jumping, maximum strength and power are linked to the CSAs of muscle fibers, especially Type IIa and IIx [1,2,3,4,14,48,57], with the reported correlations being of the same magnitude as those observed in the present study. As was also observed in the present study, the positive correlations between muscle fibers’ CSAs and muscle strength/power performances were consistently reported in a wide spectrum of individuals, ranging from non-trained to highly trained strength, power and endurance athletes, even when they were all controlled for as one group or when these correlations were performed in each group separately. Therefore, the positive impact of large muscle fibers, and especially Type IIa and IIx, is indisputable, with several scientific works explaining why large muscle fibers are crucial for sprinting, jumping, strength and power performances. However, neither the correlations reported in the literature nor the correlations observed in the present study can explain the vast majority of variations in sprinting, jumping, strength and power performances. Indeed, the correlations in the present study could only explain 23.9–48.8% of the variations, which cannot be considered high; these observations verify previous reports [1,2,3,4,14,48,57]. In contrast, no significant correlations were observed between the CSAs of each muscle fiber type and endurance performance—either when all athletes were controlled for as one group or when examined as separate groups. The reported results regarding the impact of muscle fibers’ CSA on endurance performance are very controversial. Some studies conclude that large or relatively larger muscle fibers may lead to lower or greater endurance performance; some contend that the larger Type I and the smaller Type II muscle fibers favor endurance performance; meanwhile, other studies conclude that relatively large Type II muscle fibers, at least indirectly, are crucial for endurance performance [2,6,7,9,10,16,17,18,24,30,39,56,58,61,62,63,64]. At least theoretically, relatively large, specifically trained muscle fibers, due to training-induced adaptations in their metabolic and mechanical properties, are expected to have at least some positive impact on endurance performance, mostly through the enhancement of athletes’ energy/running economy [18,35,40,44,56,61,62,64]. This can be manifested through enhanced oxidative/glycolytic metabolisms (primarily in Type I and IIa muscle fibers) or through increased biomechanical and neuromuscular efficiencies (primarily in Type IIa muscle fibers), particularly in well-trained endurance athletes, such as those in the Marathon Runners group [16,18,43,44,56,58,61,63,64]. However, according to results of the present study, and at least among its athletes, it appears that the size of each muscle fiber type does not correlate with endurance performance, either positively or negatively. This was observed both when all athletes were analyzed as a single group and when these correlations were examined separately for each group. Finally, according to the results of multiple regression analysis and LDA, it seems that the CSAs of muscle fibers are not crucial determinants of the evaluated performances; moreover, they cannot be used to distinguish successful Sprinters from Marathon Runners. Thus, considering all of the above, it could be concluded that muscle fibers’ CSAs, even if they moderately correlate with sprinting, jumping, strength and power performances, are not the muscle fiber composition parameter that could be used for the greatest explanation of human athletic performance in well-trained Sprinters and Marathon Runners. Consequently, it appears that muscle fibers’ CSAs cannot be used as distinguishing factors for the identification of Sprinters and Marathon Runners.

In contrast to muscle fibers’ percentages and CSAs, the greatest and strongest correlations were observed between Type I, IIa, IIx muscle fibers’ %CSAs and sprinting, jumping, strength and power performances, both when all athletes were controlled for as one group but also in each group separately. In all cases, the magnitude and direction of each correlation were type-dependent, with strong but negative correlations being observed between Type I %CSA and sprinting, jumping, strength and power performances, while higher and positive correlations were observed for Type IIa and IIx, with those for Type IIx being higher than those observed for Type IIa. The results of the present study provide further support to a limited number of previous studies reporting significant and strong correlations between the %CSAs of muscle fibers and sprinting, jumping, strength, power performances in sedentary and trained athletes [2,3,4], in which the magnitude of correlations was similar to that observed in the present study. All muscle fiber types seem to produce the same amount of peak force [31,65], but they exhibit significant differences between them in terms of their activation and shortening capabilities, with Type IIx having the fastest activation-action potential propagation velocities along their sarcolemma-shortening velocities, followed by Type IIa and then by Type I muscle fibers [2,66]. Therefore, muscle fibers’ power production follows the order Type IIx > Type IIa > Type I [31,32,33,35,37,57,65,66]. However, the capacities of muscle fibers for force and power production are directly related to their sizes, with individuals having larger muscle fibers (especially of Type IIa and mostly of Type IIx) being able to produce greater force and power compared to those with smaller muscle fibers [1,2,3,4,11,12,13,14,17,19,20,21,24,31,32,33,34,35,36,37,48,58,65,66], especially when specific neuromuscular adaptions are present [1,4,50], and muscle quality is high [14], as can be expected in well-trained Marathon Runners and Sprinters. Furthermore, the greatest correlations for endurance performance were observed with Type I (positive) and Type IIx (negative) %CSA compared to those observed for the distribution of these types—either when all athletes were controlled for as one group or when each group was examined separately. This was not a surprising observation, as it is well documented that individuals possessing many and large Type I muscle fibers are also characterized by increased endurance capacities, in contrast with individuals having many and large Type IIx muscle fibers [6,7,8,9,10,12,18,24,29,30,35,58,61,64,67,68]. One interesting observation of the present study was that when all athletes were examined as one group, no significant correlations were observed between Type IIa muscle fibers’ %CSA and endurance performance. However, when these correlations were examined in each group separately, significant, strong and positive correlations were observed between Type IIa muscle fibers’ %CSA and endurance performance. Thus, it may be false to state that endurance performance is not related to Type IIa %CSA or that it does not positively affect the athletes’ Type IIa %CSA, with those possessing greater Type IIa %CSA exhibiting higher endurance performance compared to those having smaller Type IIa %CSA within their group. The absence of any significant correlation between Type IIa muscle fibers’ %CSA and endurance performance, when analyzing all athletes as a single group, can be attributed to the overlapping athletes’ characteristics. As shown in Table 1 and Figure 1B,C, Sprinters exhibit a greater %CSA of Type IIa fibers compared to Marathon Runners. Conversely, Marathon Runners display significantly higher EVO_2max_ and MAV. This grouping and overlapping of data between the two groups confounded the analysis and prevented the establishment of a significant correlation when all athletes were analyzed as a single group. In contrast, only when these correlations were examined for each group separately were the “true” positive correlations between endurance performance and muscle fiber Type IIa %CSA revealed. This observation is of high importance, as it provides strong evidence regarding the positive relationship between the %CSA of Type IIa muscle fiber and endurance performance. Type IIa muscle fibers are generally considered less metabolically efficient than Type I muscle fibers [31,38,39,69,70]. However, this conclusion is derived from studies examining muscle fibers’ ATP consumption and energy efficiency during long-term maximum isometric contractions or using experimental models to simulate maximum isometric contractions in laboratory conditions on isolated muscle fibers. As the muscle fibers’ energy consumption during dynamic contractions (e.g., muscle contraction that requires faster cross-bridge turnovers than those observed during isometric contractions, such as those during running) depends on the power output (which varies depending on muscle fiber type/size), as well as on the shortening velocities, muscle contractions’ frequency and the total time of each muscle contraction, it seems that the peak energy/thermodynamic efficiency between Type I and IIa muscle fibers is similar, especially when muscle fibers’ mechanical and metabolic properties are considered relative to the size of each muscle fiber [16,31,37,39,61,62,67,71]. Thus, metabolically, Type IIa muscle fibers, during fast, dynamic contractions, are similar to Type I muscle fibers. The only differences between Type I and Type IIa muscle fibers seem to be in the time that each muscle fiber type reaches its peak efficiency, with Type IIa reaching it faster at a greater relative load and over a wider range of fast contraction velocities/frequencies compared to Type I, whose energy/thermodynamic efficiency drops dramatically as the shortening velocity increases [16,31,37,39,61,62,67,71]. Furthermore, Type IIa muscle fibers, while also oxidative, exhibit faster contraction speeds, characterized by greater capacities for power production and for anaerobic metabolism compared to Type I, which allows them to contribute significantly to performance during high-intensity running efforts [18,61,64]. However, all of the above are also affected by the type of systematic training and the training-type-dependent specific adaptations in muscle fibers’ size and metabolic properties [31,32,33,34,35,36,37,40,41,42,43,44,45], as it has been discussed in previous paragraphs. It is well established that muscle hypertrophy beyond a point is a negative factor in endurance performance [64,68,72]. This seems to also be true in the present study. Indeed, Sprinters had significantly larger Type IIa and IIx muscle fibers compared to Marathon Runners, as well as lower endurance performances. Thus, extensive hypertrophy of Type IIa and IIx muscle fibers seems to negatively affect endurance performance. However, the observed positive relationships between Type IIa %CSA and endurance performance in each group separately indicates that a relatively increased size of Type IIa muscle fibers is needed for athletes to achieve greater endurance performance, regardless of their sports specialization. Indeed, it has been suggested that relatively large Type IIa muscle fibers favor endurance performance by enhancing the neuromuscular, biomechanical and metabolic efficiency of muscles during distance running (especially at moderate and high intensities), and thus, athletes’ running economy is positively affected, which in turn leads to greater endurance capacities [18,56,58,61,64]. Therefore, it seems that endurance performance is strongly affected not only by a high Type I %CSA but also by a relatively increased %CSA of Type IIa muscle fibers.

Finally, one of the main questions of the present study was to identify which of the evaluated parameters of muscle fiber composition is stronger and consistently linked to athletes’ performances and can thus better explain athletes’ performance, as well as which of them could be used to discriminate Sprinters from Marathon Runners and vice versa. According to the results of the present study, it seems that the parameter of muscle fiber composition, which can achieve the above aim, is the %CSA of muscle occupied by Type I, IIa and IIx muscle fibers. Indeed, the results of the correlations analyses revealed that the %CSA of each muscle fiber type can explain to a greater extent the vast majority of athletes’ sprinting, jumping, strength, power and endurance performance variations (percentages of performances’ variations explained: 53.4–79.5%), while neither muscle fibers’ percentages nor CSAs alone could achieve these values. Furthermore, the results of multiple regression analysis and LDA also indicated that the %CSA of muscle fibers was the most important determinant parameter of athletes’ performance, while it was the only parameter of muscle fiber composition, which could be used to successfully discriminate Sprinters and Marathon Runners. The %CSA better represents the metabolic and mechanical properties of the whole muscle because it incorporates information regarding both the type and size of muscle fibers. For this reason, the reported correlations between either the percentage or the CSA of muscle fiber and human athletic performance can be characterized as low to moderate; they are not consistent, and they can explain only a small portion of athletes’ performance variations. Furthermore, as has been reported previously, the performance of a world class athlete involved in a power-oriented sport could only be explained by the %CSA of his muscle fiber and not only by the number and size of his Type I and II muscle fibers [13], a result, which was later verified in power-trained athletes with different training experiences [4]. In support of the above observation, it seems that the %CSA of each muscle fiber type can better explain body composition and glycemic/lipidemic profiles in sedentary individuals, while, again, it is only the %CSA of each muscle fiber type that is highly linked to training-induced adaptions in strength/power performance, body composition and glycemic/lipidemic profiles [2,3,4,46,47,73].

The present study has two limitations. Firstly, muscle fiber composition parameters were evaluated in only one muscle, specifically the vastus lateralis. Even if vastus lateralis is one of the most investigated muscles, especially in studies exploring muscle fiber composition, and it is commonly believed that vastus lateralis provides reliable and comparable information regarding the individuals’ whole body muscle fiber composition, there are differences in muscle fiber composition between muscles in each individual. Secondly, muscle fiber composition was estimated through only one muscle biopsy sampling per athlete. Recently, it has been reported that 2–3 muscle biopsy samples should be taken per individual for a more accurate estimation of muscle fiber composition [74,75]. However, in the present study, we followed the suggestion from a recent study proposing that when muscle biopsies are taken, several cuts should be performed in each incision, and the needle should be rotated 180° before the next cut [75]. Furthermore, even if biopsies were only taken from one site of the vastus lateralis, for reliable, accurate and low-variation data (2.2–6.2%) regarding each individual’s muscle fiber composition (percentage, CSA, etc.), especially for the vastus lateralis, at least 150–200 different muscle fibers should be evaluated [76,77]. To minimize this thread as much as we could, in the present study, an average of 762 different muscle fibers from each athlete were evaluated.

## 5. Conclusions

In conclusion, the results of the present study indicate that the percentage cross-sectional area of the muscle occupied by Type I, IIa and IIx muscle fibers is the most important variable of muscle fiber composition (among those evaluated in this study) for explaining and determining Sprinters’ and Marathon Runners’ sprinting, jumping, strength, power and endurance performances. It is also the parameter of muscle fiber composition that could be used to successfully distinguish Sprinters from Marathon Runners. In contrast, neither the percentage nor the cross-sectional area of muscle fibers seemed to provide sufficient information to fully understand or determine, from the perspective of muscle or muscle fibers alone, the performance of well-trained athletes. Additionally, the results of the present study suggest that neither the percentage nor the cross-sectional area of muscle fibers can reliably discriminate athletes from different sports specialization, even when they belong to the two ends of the athletic activity spectrum. Therefore, it may be time to reconsider our perspective on the significance of each of the three muscle fiber composition parameters that are evaluated in this study. Greater emphasis should be placed on the percentage area of the muscle that is occupied by each muscle fiber type in our efforts to better understand the physiological, metabolic and mechanical foundations of athletic performance.

## Figures and Tables

**Figure 1 sports-13-00074-f001:**
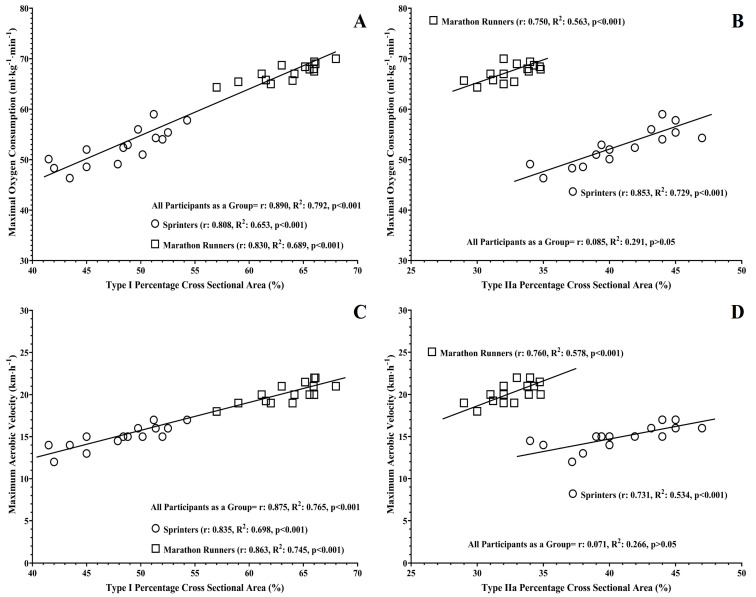
Correlation plots between vastus lateralis percentage areas occupied by Type I, IIa muscle fibers, estimated maximum oxygen consumption (**A**,**B**) and maximum aerobic velocity (**C**,**D**) for all athletes as one group (n = 30) as well as separately for Sprinters (n = 15) and Marathon Runners (n = 15).

**Table 1 sports-13-00074-t001:** Athletes’ characteristics.

		Sprinters (n = 15)	Marathon Runners (n = 15)
**Age (yrs)**		24.7 ± 4.1	26.1 ± 3.2
**Body Mass (kg)**		82.7 ± 8.8 *	69.7 ± 4.3
**Height (cm)**		180.2 ± 2.9	175.6 ± 3.4
**Body Fat (%)**		11.3 ± 4.2	9.5 ± 3.3
**Total Lean Body Mass (kg)**		67.2 ± 5.2 *	61.8 ± 2.7
**Lower Extremities Body Fat (%)**		13.6 ± 6.2	10.8 ± 2.4
**Lower Extremities Lean Body Mass (kg)**		23.6 ± 3.5 *	19.8 ± 2.9
**Vastus Lateralis Muscle Fiber Composition**			
**Percentage**	**Type I (%)**	54.2 ± 6.8	61.9 ± 5.4 *
	**Type IIa (%)**	35.8 ± 2.5	35.9 ± 5.5
	**Type IIx (%)**	9.2 ± 3.1 *	2.2 ± 1.3
	**Type II (%)**	44.8 ± 5.3	38.1 ± 4.4
**Cross-Sectional** **Area**	**Type I (μm^2^)**	4754 ± 574	4269 ± 423
	**Type IIa (μm^2^)**	6201 ± 654 *	4824 ± 847
	**Type IIx (μm^2^)**	5242 ± 484 *	3010 ± 599
**Percentage Cross-Sectional Area**	**Type I (%)**	48.6 ± 5.6	63.8 ± 3.2 *
	**Type IIa (%)**	41.2 ± 5.7 *	33.5 ± 2.2
	**Type IIx (%)**	10.2 ± 3.9 *	2.7 ± 0.8
	**Type II (%)**	51.4 ± 6.1 *	35.9 ± 4.2
**Performance Parameters**			
**Sprinting**	**10 m Time (sec.)**	1.5 ± 0.2 *	1.9 ± 0.2
	**10 m Velocity (m·sec.^−1^)**	6.5 ± 0.3 *	5.1 ± 0.3
	**60 m Time (Sec.)**	6.9 ± 0.2 *	8.7 ± 0.2
	**60 m Velocity (m·sec.^−1^)**	8.7 ± 0.2 *	7.1 ± 0.3
**Counter Movement Jump**	**Height (cm)**	46.2 ± 2.3 *	29.1 ± 0.8
	**Power (W)**	1781.2 ± 398.1 *	799.4 ± 234.2
**Isometric Leg Press**	**Maximum Isometric Force (N)**	4892.5 ± 1271.6 *	3146.1 ± 800.1
	**Rate of Force Development at 60 ms (N·Sec.^−1^)**	17,632.3 ± 5698.4 *	6480.5 ± 3905.1
	**Rate of Force Development at 120 ms (N·Sec.^−1^)**	20,557.2 ± 2419.5 *	10,247.7 ± 2148.7
	**Rate of Force Development at 250 ms (N·Sec.^−1^)**	15,178.3 ± 1802.5 *	8026.8 ± 871.3
**Aerobic** **Capacity**	**Estimated Maximal Oxygen Consumption (mL·kg^−1^·min^−1^)**	52.2 ± 4.8	67.5 ± 3.8 *
	**Maximum Aerobic Velocity (km·h^−1^)**	14.9 ± 1.8	20.2 ± 1.4 *

(*) denotes significant differences between groups, with the marked group achieving the greatest values or performances (*p* < 0.05).

**Table 2 sports-13-00074-t002:** Correlations between vastus lateralis muscle fiber parameters and athletes’ sprinting, jumping, strength, power and endurance performances, for all athletes as a group (n = 30; only significant correlations are presented).

Performance Parameter	Percentage			Cross-Sectional Area			Percentage Cross-Sectional Area		
	I	IIa	IIx	I	IIa	IIx	I	IIa	IIx
**10 m Sprint Time**					−0.589 ^#^	−0.614 ^#^	0.842 ^#^	−0.800 ^#^	−0.895 ^#^
**10 m Sprint Velocity**				0.537 *	0.600 ^#^	0.652 ^#^	−0.789 ^#^	0.829 ^#^	0.865 ^#^
**60 m Sprint Time**					−0.666 ^#^	−0.628 ^#^	0.836 ^#^	−0.811 ^#^	−0.889 ^#^
**60 m Sprint Velocity**				0.566 ^#^	0.689 ^#^	0.691 ^#^	−0.806 ^#^	0.869 ^#^	0.872 ^#^
**CMJ Height**	−0.481 *	0.512 *	0.521 *	0.516 *	0.669 ^#^	0.699 ^#^	−0.848 ^#^	0.828 ^#^	0.892 ^#^
**CMJ Power**	−0.500 *	0.589 *	0.536 *	0.528 *	0.639 ^#^	0.700 ^#^	−0.789 ^#^	0.814 ^#^	0.878 ^#^
**LPMIF**	−0.514 *	0.441 *	0.478 *	0.599 ^#^	0.615 ^#^	0.636 ^#^	−0.771 ^#^	0.760 ^#^	0.788 ^#^
**LPRFD_60_**	−0.433 *		0.476 *	0.489 *	0.668 ^#^	0.692 ^#^	−0.701 ^#^	0.845 ^#^	0.869 ^#^
**LPRFD_120_**	−0.465 *		0.498 *	0.584 ^#^	0.689 ^#^	0.697 ^#^	−0.712 ^#^	0.887 ^#^	0.902 ^#^
**LPRFD_250_**	−0.478 *		0.485 *	0.601 ^#^	0.692	0.645 ^#^	−0.753 ^#^	0.891 ^#^	0.841 ^#^
**EVO_2max_**	0.477 *		−0.489 *				0.890 ^#^		−0.812 ^#^
**MAV**	0.505 *		−0.502 *				0.875 ^#^		−0.824 ^#^

(*) denotes *p* < 0.05, (^#^) denotes *p* < 0.005. CMJ: Counter Movement Jump; LPMIF: Leg Press Maximum Isometric Force; LPRFD_xx_: Leg Press Rate of Force Development at xxms from the onset of muscle contraction; EVO_2max_: Estimated Maximal Oxygen Consumption; MAV: Maximum Aerobic Velocity.

**Table 3 sports-13-00074-t003:** Beta coefficients (B) as indicators of muscle fibers’ composition parameters’ relative strengths/impact on athletes’ sprinting, jumping, maximum isometric force and rate of force development performances when all athletes were controlled for as one group (only significant coefficients are presented, n = 30).

		10 m Sprint Time	10 m Sprint Velocity	60 m Sprint Time	60 m Sprint Velocity	CMJ Height	CMJ Power	LPMIF	LPRFD_60_	LPRFD_120_	LPRFD_250_
**R^2^/p** **(Variance inflation factor** **/Tolerance)**		0.896/<0.001 (1.612–3.974/0.252–0.514)	0.876/<0.001 (1.302–4.850/0.315–0.602)	0.933/<0.001 (2.930–3.491/0.245–0.465)	0.941/<0.001 (2.930–3.491/0.285–0.602)	0.815/<0.001 (1.015–3.298/0.303–0.698)	0.817/<0.001 (2.061–4.974/0.485–0.792)	0.796/<0.001 (2.111–4.537/0.394–0.824)	0.801/<0.001 (1.198–3.259/0.394–0.932)	0.978/<0.001 (2.107–4.449/0.425–0.810)	0.949/<0.001 (2.298–3.298/0.303–0.749)
**Percentage or Cross Sectional** **Area**	**I**										
	**IIa**										
	**IIx**										
**Percentage Cross Sectional** **Area**	**I**	0.652	−0.593	0.362	−0.372	−0.666	−0.418	−0.678	−0.748	−0.849	−0.549
	**IIa**	−0.715	0.493	−0.293	0.431	0.723	0.625	0.715	0.643	0.589	0.782
	**IIx**	−0.829	0.685	−0.641	0.635	0.815	0.893		0.899	0.925	0.471

CMJ: Counter Movement Jump; LPMIF: Leg Press Maximum Isometric Force; LPRFD_xx_: Leg Press Rate of Force Development at xxms from the onset of muscle contraction.

**Table 4 sports-13-00074-t004:** Beta coefficients (B) as indicators of muscle fibers’ composition parameters’ relative strengths/impact on athletes’ endurance performances when each group was examined separately (only significant coefficients are presented, n = 15/group).

		Sprinters		Marathon Runners	
		Estimated Maximal Oxygen Consumption	Maximum Aerobic Velocity	Estimated Maximal Oxygen Consumption	Maximum Aerobic Velocity
**R^2^/p** **(Variance inflation factor** **/Tolerance)**		0.796/<0.001 (2.217–4.870/0.623–0.914)	0.888/<0.001 (2.451–4.666/0.289–0.519)	0.989/<0.001 (1.9894.465/0.523–0.824)	0.842/<0.001 (1.874–3.335/0.385–0.701)
**Percentage or Cross Sectional Area**	**I**				
	**IIa**				
	**IIx**				
**Percentage Cross Sectional** **Area**	**I**	0.785	0.625	0.985	0.971
	**IIa**	0.666	0.528	0.824	0.749
	**IIx**	−0.689	−0.701	−0.279	−0.389

**Table 5 sports-13-00074-t005:** Results of the linear discriminant analysis.

		Canonical Discriminant Function Coefficients	Wilks’ Lambda	*p* Value
**Percentage**	**I**	0.287	0.748	0.045
	**IIa**	0.209	0.849	0.125
	**IIx**	−0.121	0.997	0.287
**Cross-Sectional Area**	**I**	0.001	0.699	0.039
	**IIa**	−0.002	0.724	0.048
	**IIx**	−0.002	0.811	0.121
**Percentage Cross-Sectional Area**	**I**	**0.752**	**0.308**	**<0.001**
	**IIa**	**0.663**	**0.361**	**<0.001**
	**IIx**	**1.185**	**0.289**	**<0.001**

The strongest contributors are marked with bold letters, while non-significant contributors are denoted with gray letters.

## Data Availability

Data are available upon reasonable request from the authors.

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
