# Peer review of "Sprinters’ and Marathon Runners’ Performances Are Better Explained by Muscle Fibers’ Percentage Cross-Sectional Area than Any Other Parameter of Muscle Fiber Composition"

_sports, 2025, doi:10.3390/sports13030074_

Round 1
Reviewer 1 Report
Comments and Suggestions for Authors
Thank you inviting me to review the paper aimed at investigate the correlation between muscle fiber type variables and sprinting, jumping, strength, power and endurance performances in sprinters and marathon runners.
The manuscript is well written in general.
Major points:
I do have a concern with these types of polarized comparisons in terms of subject cohorts as the vast difference in training history contributes to the differences in performance outcomes, which may mask the strong associations found with muscle fibre composition. For example, if you recruit sprint/power athletes they train very differently to endurance athletes, and there are also likely differences in muscle fibre composition. Is it the difference in training history or muscle fibre composition that is responsible for the differences in performance outcomes?
A lot of the literature assessing muscle fibre composition and performance outcomes may be clouded by the poor reliability of determining muscle fibre composition from biopsies (Sahl et al. 2018; Horwath et al. 2021; Van de Casteele et al. 2024). This should be discussed in the manuscript.
Introduction
Line 38 - It would also be worthwhile citing a paper that has looked at the estimated muscle fibre composition in relation to vertical (jumping) and horizontal (sprinting) force velocity profiles as these are directly relevant to the manuscript - Relationships between lower limb muscle characteristics and force-velocity profiles derived during sprinting and jumping - Medicine and Science in Sports and Exercise 53 (7), 1400-1411
Overall, the introduction provides a nice overview of a complex area assessing the association between muscle fibre composition and performance variables across sprint and endurance disciplines
Line 119 – these findings are probably due to the nature of the training history of the groups, rather than just muscle fibre composition. This also related to the design of your study, how can you be sure the association between muscle fibre composition and performance variables are due to the muscle fibre composition or due to the differences in training history?
Methods
The methods section is clear and all procedures are described in good detail.
Statistical analysis – You quantified lean mass with the DXA scan so why did you not include this as a covariate and run a partial correlation with all of your analyses? This could control for the large influence of lean mass, and thus do a better job of isolating the influence/association between muscle fibre composition and the performance parameters.
Discussion
Line 471 – This is because muscle fibre composition is not the only factor relevant for performance. Of course, overall muscle volume is important (which you can control for) and muscle architecture etc. As such, to expect almost performance correlations is not appropriate or justified.
Line 532 – This relates to my major concern point above whereby the true association between muscle fibre composition and performance variables is obfuscated by training history when you compared to polarized cohorts who have many years of polarized training
Author Response
Authors Responses to Reviewer 1 Comments
General Comment
Thank you inviting me to review the paper aimed at investigate the correlation between muscle fiber type variables and sprinting, jumping, strength, power and endurance performances in sprinters and marathon runners.
The manuscript is well written in general.
Authors’ Response:
We would like to thank the reviewer for the time spent on this manuscript, the valuable comments and suggestions. We have now revised our manuscript according to the reviewer’s suggestions. Below, you can find our point-by-point responses.
Major points:
Reviewer Comments
I do have a concern with these types of polarized comparisons in terms of subject cohorts as the vast difference in training history contributes to the differences in performance outcomes, which may mask the strong associations found with muscle fibre composition. For example, if you recruit sprint/power athletes they train very differently to endurance athletes, and there are also likely differences in muscle fibre composition. Is it the difference in training history or muscle fibre composition that is responsible for the differences in performance outcomes?
Line 119 – these findings are probably due to the nature of the training history of the groups, rather than just muscle fibre composition. This also related to the design of your study, how can you be sure the association between muscle fibre composition and performance variables are due to the muscle fibre composition or due to the differences in training history?
Line 532 – This relates to my major concern point above whereby the true association between muscle fibre composition and performance variables is obfuscated by training history when you compared to polarized cohorts who have many years of polarized training
Authors’ Response:
We would like to response at once in the above comments from the reviewer, as it seems that are related. We totally agree with reviewers’ comments. In fact, the results of the present study provide strong evidence about this issue, indicating that it is the training induced adaptations on %CSA of muscle fibers that is responsible for the differences in performance outcomes. As the reviewer has already read, all the results of the present study indicate that the %CSA of muscle fibers seems to be the determinant parameter that could better explain the performance differences between Sprinters and Marathon Runners, and not the percentage or the cross-sectional area of each muscle fiber Type that these athletes possessed. Similarly, the %CSA of muscle occupied by Type I, IIa and IIx muscle fibers was the only parameter that could discriminate Sprinters from Marathon Runners, and not any other variable of muscle fiber composition. As we have mentioned in the present manuscript, the %CSA of muscle fibers, is mostly affected by the training history, and not so much from the genetic background of each athlete (as mentioned in the text, lines 95-128). Thus, the %CSA of muscle fibers, in contrast to the percentages or the cross-sectional area of muscle fibers alone, provides information about the physiological, mechanical and metabolic properties of muscles at a given time point, as well as for the specific training induced adaptations on the above parameters, as the outcomes of athletes specific and different training background. This was the reason why we recruited athletes with totally different training backgrounds. In an effort to be clearer to readers, and taking in advance the comment of the reviewer, we added the following in the introduction section (Lines 119-128):
“As muscle fibers and muscles are characterized by high plasticity (the ability to adapt and change depending on the imposed physiological, mechanical and metabolic demands of each period), their morphological characteristics, muscle fiber distribution, cross-sectional area, physiological, metabolic and mechanical properties are not something stable but they change according to the imposed demands, with systematic training to be one of the most important regulator of muscle fibers’ and muscles’ phenotypes [40,42-46]. The epigenetic – training induced changes on muscle fibers and whole muscle phenotype, seems to better be explained by each individual/athlete muscle fibers %CSA at a given time point, compared to any other parameter of muscle fiber composition, as previously described [2,3,13].”
Reviewer Comment
A lot of the literature assessing muscle fibre composition and performance outcomes may be clouded by the poor reliability of determining muscle fibre composition from biopsies (Sahl et al. 2018; Horwath et al. 2021; Van de Casteele et al. 2024). This should be discussed in the manuscript.
Authors’ Response:
Thank you for pointing out this. We have now added the following in our revised manuscript (Lines 633-649):
“The present study has two limitations. Firstly, muscle fiber composition parameters were evaluated in only one muscle, specifically from vastus lateralis. Even if vastus lateralis is one of the most investigated muscles, especially in studies exploring muscle fiber composition, and it commonly believed that vastus lateralis provides reliable and comparable information about individuals’ whole body muscle fiber composition, there are differences in muscle fiber composition between muscles in each individual. Secondly, muscle fiber composition was estimated through only one muscle biopsy sampling per athlete. Recently, it has been reported that 2-3 muscle biopsies samples should be taken per individual for a more accurate estimation of muscle fiber composition [75,76]. However, in the present study we followed the suggestion of a recent study, proposing that when muscle biopsies are taken, several cutting should be done in each incision, and the needle should be rotated 180o before the next cut [76]. Furthermore, even if biopsies were taken only from one site of vastus lateralis, for reliable, accurate and with low variation data (2.2-6.2%) about each individuals’ muscle fiber composition (percentage, CSA etc), especially for vastus lateralis, at least 150-200 different muscle fibers should be evaluated [77,78]. To minimize this thread as much as we could, in the present study, an average of 762 different muscle fibers from each athlete were evaluated.”
Reviewer Comment
Introduction
Line 38 - It would also be worthwhile citing a paper that has looked at the estimated muscle fibre composition in relation to vertical (jumping) and horizontal (sprinting) force velocity profiles as these are directly relevant to the manuscript - Relationships between lower limb muscle characteristics and force-velocity profiles derived during sprinting and jumping - Medicine and Science in Sports and Exercise 53 (7), 1400-1411
Authors’ Response:
Thank you for this suggestion. We have now added the suggested ref in our revised manuscript
Reviewer Comment
Overall, the introduction provides a nice overview of a complex area assessing the association between muscle fibre composition and performance variables across sprint and endurance disciplines
Methods
The methods section is clear and all procedures are described in good detail.
Authors’ Response:
Thank you very much for the above comments, they are much appreciated.
Reviewer Comment
Statistical analysis – You quantified lean mass with the DXA scan so why did you not include this as a covariate and run a partial correlation with all of your analyses? This could control for the large influence of lean mass, and thus do a better job of isolating the influence/association between muscle fibre composition and the performance parameters.
Authors’ Response:
We have already performed all the analyses, with lean mass as covariance. The magnitude of the correlations was not affected by the effect of lean mass (Pearson r range: -0.848 to 0.902; partial correlations with muscle mass as covariance: -0.825 to 0.895). Thus, considering the above, as well as that lean mass is determined primarily from the cross-sectional area of muscle fibers (a variable that examined and presented thoroughly in the present study), and that the focus of the present study was on muscle fiber composition, we decided not to include this information in our manuscript. However, taking into advance the comment of the reviewer, we have now added this information to our revised manuscript:
Lines 278-279
“Partial correlations used to investigate the possible effect of lean mass on the above correlations”
Lines 346-348
“Finally, when the above correlations were controlled for the possible effect of athletes’ lean mass, the magnitude of the correlations were not affected significantly (range: -0825 to 0.895).”
Reviewer Comment
Discussion
Line 471 – This is because muscle fibre composition is not the only factor relevant for performance. Of course, overall muscle volume is important (which you can control for) and muscle architecture etc. As such, to expect almost performance correlations is not appropriate or justified.
Authors’ Response:
We totally agree with this. Overall athletes’ performance is determined by many factors, including the function of neural system, muscle mass-architecture, muscle fiber composition, metabolic, mechanical and physiological properties of muscle fibers and muscle and many more. However, in the present study we focused on the impact of muscle fiber, percentages, cross-sectional area and %CSA only. Furthermore, in this part of the manuscript we are discussing only the correlations between athletes’ performance and cross-sectional area. According to the results of the present study, among the parameters of muscle fiber composition that this study investigates, muscle fibers’ CSA is not a parameter that alone could explain athletes’ performance. And this is what we are state in this part of the manuscript.
Again, we would like to thank the reviewer for the time spent on our manuscript, and the valuable comments, which we believe that they was to the point and they help us to make our manuscript better

Reviewer 2 Report
Comments and Suggestions for Authors
I’ve got the opportunity to be a reviewer of the paper „Sprinters’ and Marathon Runners’ Performances are Better Explained by Muscle Fibers’ Percentage Cross Sectional Area than any Other Parameter of Muscle Fiber Composition”. This is an interesting piece of work with several practical findings for sports science. However, I also noted some areas that must be improved. Please find detailed points below.
Major comments:
- specify the location and dates of recruitment. List all the sports clubs mentioned in line 143.
- Did you reach significance for the power of your sample? Did you do any power analysis?
- How did you evaluate participants’ training background to assign them into 2 groups?
- Which equation did you use to estimate VO2max? Remember that VO2max models could present a wide error of about 20 mL/kg/min in athletes.
- When do you measure body composition? In the morning, in the fasted state, or in another part of the day?
- Specify inclusion and exclusion criteria more precisely.
- Did you evaluate only 30 participants or more, and some were excluded? If yes, list how many athletes you excluded.
- You applied the interclass correlation coefficient, which has 3 types. Which one did you use?
- For R2 and, in general, the regression analysis you should present also beta (B). Please add this parameter to your tables.
Minor comments:
- provide R2 for your regressions in the results for the abstract.
- stratify the introduction into smaller parts of text to facilitate reading.
- In line 38 you cite 22 papers in one place, which is far too much for one claim. Keep only 1-2 references here. The similar situation occurred in several places in the text. Support one claim with 1-2 (maximum 3) references and no more.
- Reduce the amount of text in the introduction and make the introduction as a brief 1-pager.
- Remember that key papers about the principles of running performance should be discussed. See doi: 10.3390/jcm12082884 and doi: 10.1038/s41598-024-73730-z.
- Start your discussion into smaller parts of the text similarly to the introduction.
- The discussion could also be shorter.
To sum up my report, the authors should apply revisions. They should especially focus on major comments.
Author Response
Authors Responses to Reviewer 2 Comments
General Comment
I’ve got the opportunity to be a reviewer of the paper „Sprinters’ and Marathon Runners’ Performances are Better Explained by Muscle Fibers’ Percentage Cross Sectional Area than any Other Parameter of Muscle Fiber Composition”. This is an interesting piece of work with several practical findings for sports science. However, I also noted some areas that must be improved. Please find detailed points below.
Authors’ Response:
We would like to thank the reviewer for the time spent on this manuscript, the valuable comments and suggestions. We have now revised our manuscript according to the reviewer’s suggestions. Below, you can find our point-by-point responses.
Major points:
Reviewer Comments
- specify the location and dates of recruitment. List all the sports clubs mentioned in line 143.
Authors’ Response:
Dear reviewer, this is something that we can no do, because if we provide these information we violate one of the strict rules of our Ethics Committee, which is not to published any information that could leads to the identity of our participants, while all participants signed an informed consent which stated that no information which could revealed who they are or any information which may lead to their identity will be published. This is also a rule of Sports Journal. Thus, in respect to reviewer comment, this is something that is forbidden from us to do
Reviewer Comments
- Did you reach significance for the power of your sample? Did you do any power analysis?
Authors’ Response:
Thank you for pointing this out. We have done a post-hoc power analysis. We have now included this information in our revised manuscript.
Lines 265-268
“A post-hoc power analysis was used for the determination of the results’ actual power (G*Power ver. 3.1; Frank Faul, Universitat Kiel, Germany). The power for the correlation coefficients when all participants were included (n=30) ranged between 0.875 and 0.902, while for all the other analyses ranged between 0.850 and 0.898.”
Reviewer Comments
- How did you evaluate participants’ training background to assign them into 2 groups?
Authors’ Response:
As the reviewer has already read in the initially submitted manuscript, all participants were well-trained athletes, who trained for their events for at least five years and compete in national championships in their events. Thus, each athlete was assigned into the two groups based on the event that participated in the national championships and according to the official records of our national federation. Furthermore, all participants completed before the initiation of the study, a recall self-reported physical activity questionnaire, which has been used in several studies of our lab until now [1-20]. This questionnaire provides us all the needed information about the training background and the training that each athlete followed until the initiation of the study.
Reviewer Comments
- Which equation did you use to estimate VO2max? Remember that VO2max models could present a wide error of about 20 mL/kg/min in athletes.
Authors’ Response:
The equation that we used for the estimation of VO2max is already stated in the initial submitted manuscript (Lines in the initial submission: 236-237; Lines in the revised manuscript: 243-244): “….using the equation [21]: VO2peak (ml∙kg-1∙min-1): (max heart rate ∙ heart rate rest-1) ∙ 15.”. Indeed, equations for the estimation of VO2max could provide an error of 20 mL/kg/min in athletes. However, the used equation has been developed in a sample of well-trained athletes, and its reliability, reproductivity and accuracy has been examined before, while it has been used in over 200 studies until now. Thus, even if it estimates VO2max, this equation provides data very close to those observed during a test with breath-by breath analysis system. Finally, the use of the VO2max testing in the present study was not aimed to provide information or description about the actual VO2max of our athletes, but to be used for the determination of their endurance capacity. Thus, even in the unlikely event that this equation provides an error in the estimation of our athletes VO2max, this error should be constant, indicating an error of Type II. Thus, this error should be presented in all participants, and will not have any negative impact on the results of the present study (correlations, MR, LDA).
Reviewer Comments
- When do you measure body composition? In the morning, in the fasted state, or in another part of the day?
Authors’ Response:
Thank you for pointing out this. We have forgotten to add this information in our initial submission. We have now included it in the revised manuscript
Lines 184-185
“….during morning hours (8-10am) and after overnight fasting.”
Reviewer Comments
- Specify inclusion and exclusion criteria more precisely.
Authors’ Response:
All the inclusion criteria that were adapted in the present study are presented analytically in the manuscript
Reviewer Comments
- Did you evaluate only 30 participants or more, and some were excluded? If yes, list how many athletes you excluded.
Authors’ Response:
Yes 30 athletes who fulfilled the inclusion criteria starts in the present study and all completed it. No one withdraw from the study, neither any participant who fulfilled the inclusion criteria was excluded as we mention in the above comment.
Reviewer Comments
- You applied the interclass correlation coefficient, which has 3 types. Which one did you use?
Authors’ Response:
We have used the Intrarater reliability. We have now added the following in the revised manuscript
Line 188
“….correlation coefficients (ICC; Intrarater) for….”
Reviewer Comments
- For R2 and, in general, the regression analysis you should present also beta (B). Please add this parameter to your tables.
Authors’ Response:
All values of R2 and all the beta (B) values are already presented both in the text and Tables 3 and 4.
Minor comments:
Reviewer Comments
- provide R2 for your regressions in the results for the abstract.
Authors’ Response:
Thank you for pointing this out. We have now added the requested information in the abstract
Lines 27-28
“…(R2: 0.796-0.978; p < 0.001; B coefficients: -0.293 – 0.985 )…,”
Reviewer Comments
- stratify the introduction into smaller parts of text to facilitate reading.
Authors’ Response:
We totally understand the comment of the reviewer. However, during the manuscript preparation we tried to split the introduction section into smaller parts, but the meaning of our text included in the introduction section was lost, and in many cases we have to repeat the provided information. Thus, with respect to reviewer’s suggestion, we would like to keep the introduction as it is.
Reviewer Comments
- In line 38 you cite 22 papers in one place, which is far too much for one claim. Keep only 1-2 references here. The similar situation occurred in several places in the text. Support one claim with 1-2 (maximum 3) references and no more.
Authors’ Response:
We totally understand the comment of the reviewer. However, as this study investigates the impact of 3 parameters of muscle fiber composition, in several performance parameters, we have to include several studies as references, since each referred study reports some results, either for only one of the parameters of muscle fiber composition or for only one performance parameter. So by keeping only 1-3 refs per sentence, it will either leads to important references to be excluded, or the length of our manuscript to be increased proportionally. For example, in the specific part of the manuscript that the reviewer pointed out, and in many more, we have summarized in one sentence all the conclusion/results of the published papers in the area, so that readers to be able to track all these references easily and have all the needed references in this area.
Reviewer Comments
- Reduce the amount of text in the introduction and make the introduction as a brief 1-pager.
- Start your discussion into smaller parts of the text similarly to the introduction.
- The discussion could also be shorter.
Authors’ Response:
We would like to respond at once to the above three comments of the reviewer, as they link to each other. As we have responded to the previous comment of the reviewer, the nature of the present study is that, which do not allow us to have either a small (1paged introduction) or a smaller discussion. The present study did not investigate the effect of only one parameter of muscle fiber composition to one performance parameter and in one group of athletes, but investigated and compared the effects of three different parameters of muscle fiber composition in several performance parameters in two groups of totally different trained athletes. Therefore, an extensive literature review is needed, to provide to the readers all the needed information about the existent problem and the knowledge gap that exist in the literature until now and the rational of our study. Finally, the discussion section cannot be shorter, as we have made several analyses, and provide several new data-results, which, all of them, have to be discussed.. Thus, again with respect of reviewer’s comment, we would like to keep both the introduction and discussion section as it is, in an effort to provide all the needed information to the readers and explain why we have done this study and how our findings bridge the gap between theory and practice.
Reviewer Comments
- Remember that key papers about the principles of running performance should be discussed. See doi: 10.3390/jcm12082884 and doi: 10.1038/s41598-024-73730-z.
Authors’ Response:
Thank you very much for pointing us these very good studies. However, after a thorough study of both papers, we do not understand which is their relevance with this study. Both studies have no investigate or provide any result or data related to the correlations or even more to the effect of any parameter of muscle fiber composition on endurance performance. The first one (10.3390/jcm12082884), aimed to externally validate HRmax prediction models in endurance athletes independently for running and cycling CPET, and the second one (10.1038/s41598-024-73730-z) aimed to revise and adapt the standard reference equation for O2Ppeak to the endurance athlete population. Thus, both studies are totally unrelated to the aim, scope of the present study or to what parameters the present study investigates or in general in any data that included in the present study. According to the above, it is unethical to include them in the present study as they are not related to any aspect of the present study. Finally, all the key studies that investigate the effect of muscle fibers and in general of muscle physiological, metabolic and mechanical properties on endurance performance, have been discussed thoroughly, as the reviewer pointed out in a previous comment
Reviewer Comments
To sum up my report, the authors should apply revisions. They should especially focus on major comments.
Authors’ Response:
Again we would like to thank the reviewer for the time spent on our manuscript, and the valuable comments, which we believe that they was to the point and they help us to make our manuscript better
- Kostikiadis, N.I.; Methenitis, S.; Tsoukos, A.; Veligekas, P.; Terzis, G.; Bogdanis, C.G. The effect of short-term sport-specific strength and conditioning training on physical fitness of well-trained mixed martial arts athletes. Journal of Sports Science and Medicine 2018, 17, 348-358
- Kyriazis, T.; Methenitis, S.; Zaras, N.; Stasinaki, A.-N.; Karampatsos, G.; Georgiadis, G.; Terzis, G. Effects of Complex Vs. Compound Training on Competitive Throwing Performance. Journal of Strength and Conditioning Research 2022, 36, 1866-1874, doi:10.1519/JSC.0000000000004264.
- Lagodimos, P.; Thomakos, P.; Methenitis, S.; Paizis, C. The effect of weighted vest use during in-season, small-sided games training on young soccer players’ performance. Journal of Sports Sciences 2024, 42, 993-1001, doi:https://doi.org/10.1080/02640414.2024.2378267.
- Methenitis, S.; Karandreas, N.; Spengos, K.; Zaras, N.; Stasinaki, A.N.; Terzis, G. Muscle Fiber Conduction Velocity, Muscle Fiber Composition, and Power Performance. Medicine & Science in Sports & Exercise 2016, 48, 1761-1771, doi:10.1249/mss.0000000000000954.
- Methenitis, S.; Mpampoulis, T.; Spiliopoulou, P.; Papadimas, G.; Papadopoulos, C.; Chalari, E.; Evangelidou, E.; Stasinaki, A.N.; Nomikos, T.; Terzis, G. Muscle fiber composition, jumping performance and rate of force development adaptations induced by different power training volumes in females. Applied Physiology, Nutrition, and Metabolism 2020, 45, 996-1006, doi:10.1139/apnm-2019-0786.
- Methenitis, S.; Nomikos, T.; Kontou, E.; Kiourelli, K.M.; Papadimas, G.; Papadopoulos, C.; Terzis, G. Skeletal muscle fiber composition may modify the effect of nutrition on body composition, in young females. Nutrition, Metabolism and Cardiovascular Diseases 2023, 10.1016/j.numecd.2022.12.027, doi:10.1016/j.numecd.2022.12.027.
- Methenitis, S.; Nomikos, T.; Mpampoulis, T.; Kontou, E.; Evangelidou, E.; Papadopoulos, C.; Papadimas, G.; Terzis, G. Type IIx muscle fibers are related to poor body composition, glycemic and lipidemic blood profiles in young females: the protective role of type I and IIa muscle fibers. European Journal of Applied Physiology 2024, 124, 585-594, doi:10.1007/s00421-023-05302-4.
- Methenitis, S.; Nomikos, T.; Mpampoulis, T.; Kontou, E.; Kiourelli, K.M.; Evangelidou, E.; Papadopoulos, C.; Papadimas, G.; Terzis, G. Different eccentric based Power Training volumes improve glycemic, lipidemic profile and body composition of females in a dose-dependent manner. Associations with muscle fibers composition adaptations. European Journal of Sport Science 2022, 10.1080/17461391.2022.2027024, 1-28, doi:10.1080/17461391.2022.2027024.
- Methenitis, S.; Spengos, K.; Zaras, N.; Stasinaki, A.N.; Papadimas, G.; Karampatsos, G.; Arnaoutis, G.; Terzis, G. Fiber Type Composition And Rate Of Force Development In Endurance And Resistance Trained Individuals. The Journal of Strength & Conditioning Research 2019, 33, 2388–2397, doi:10.1519/jsc.0000000000002150.
- Methenitis, S.; Terzis, G.; Zaras, N.; Stasinaki, A.N.; Karandreas, N. Intramuscular Fiber Conduction Velocity, Isometric Force and Explosive Performance. J Hum Kinet 2016, 51, 93-101, doi:doi: 10.1515/hukin-2015-0174.
- Methenitis, S.; Theodorou, A.A.; Chatzinikolaou, P.N.; Margaritelis, N.V.; Nikolaidis, M.G.; Paschalis, V. The effects of chronic concentric and eccentric training on position sense and joint reaction angle of the knee extensors. European Journal of Sport Science 2023, 10.1080/17461391.2023.2184726, 1-29, doi:10.1080/17461391.2023.2184726.
- Methenitis, S.; Zaras, N.; Spengos, K.; Stasinaki, A.N.; Karampatsos, G.; Georgiadis, G.; Terzis, G. Role of Muscle Morphology in Jumping, Sprinting, and Throwing Performance in Participants With Different Power Training Duration Experience. The Journal of Strength & Conditioning Research 2016, 30, 807-817, doi:10.1519/jsc.0000000000001147.
- Mpampoulis, T.; Methenitis, S.; Papadopoulos, C.; Papadimas, G.; Spiliopoulou, P.; Stasinaki, A.N.; Bogdanis, G.; Karampatsos, G.; Terzis, G. Weak Association between Vastus Lateralis Muscle Fiber Composition and Fascicle Length in Young Untrained Females. Sports 2021, 9, 56, doi:10.3390/sports9050056.
- Mpampoulis, T.; Methenitis, S.; Stasinaki, A.N.; Zaras, N.; Bogdanis, G.; Terzis, G. Effect of Different Reduced Training Frequencies After 12 Weeks of Concurrent Ballistic and Aerobic Training on Muscle Power and Triceps Brachii Muscle Architecture. J. Funct. Morphol. Kinesiol. 2025, 10, 37, doi:10.3390/jfmk10010037.
- Mpampoulis, T.; Stasinaki, A.N.; Methenitis, S.; Zaras, N.; Bogdanis, G.; Terzis, G. Effect of Different Reduced Training Frequencies after 12 Weeks of Concurrent Resistance and Aerobic Training on Muscle Strength and Morphology. Sports 2024, 12, 198, doi:https://doi.org/10.3390/sports12070198.
- Simitzi, V.; Tsoukos, A.; Kostikiadis, I.; Parotsidis, C.; Paizis, C.; Nassis, G.; Methenitis, S. The acute effects of different high-intensity conditioning activities on sprint performance differ between sprinters of different strength and power characteristics. Kinesiology 2021, 53, 193-205, doi:10.26582/k.53.2.2.
- Spiliopoulou, P.; Zaras, N.; Methenitis, S.; Papadimas, G.; Papadopoulos, C.; Bogdanis, G.; Terzis, G. The effect of concurrent power training and high intensity interval cycling on muscle morphology and performance. Journal of Strength and Conditioning Research 2021, 35, 2464-2471, doi:10.1519/JSC.0000000000003172.
- Stasinaki, A.N.; Zaras, N.; Methenitis, S.; Tsitkanou, S.; Krase, A.; Kavvoura, A.; Terzis, G. Triceps Brachii Muscle Strength and Architectural Adaptations with Resistance Training Exercises at Short or Long Fascicle Length. Journal of Functional Morphology and Kinesiology 2018, 3, 28, doi:https://doi.org/10.3390/jfmk3020028.
- Stasinaki, A.N.; Zaras, Ν.; Methenitis, S.; Bogdanis, G.; Terzis, G. Rate of force development and muscle architecture after fast and slow velocity eccentric training. Sports 2019, 7, 41, doi:https://doi.org/10.3390/sports7020041
- Terzis, G.; Spengos, K.; Methenitis, S.; Aagaard, P.; Karandreas, N.; Bogdanis, G. Early phase interference between low-intensity running and power training in moderately trained females. European Journal of Applied Physiology 2016, 116, 1063-1073, doi:10.1007/s00421-016-3369-z.
- Uth, N.; Sørensen, H.; Overgaard, K.; Pedersen, P.K. Estimation of VO2max from the ratio between HRmax and HRrest–the heart rate ratio method. European Journal of Applied Physiology 2004, 91, 111-115, doi:https://doi.org/10.1007/s00421-003-0988-y.
Round 2
Reviewer 1 Report
Comments and Suggestions for Authors
Thank you, my comments have been addressed.
Reviewer 2 Report
Comments and Suggestions for Authors
The authors revised their paper properly.